# Multi-level Contrastive Learning for Script-based Character Understanding

**Dawei Li[1], Hengyuan Zhang[2], Yanran Li[3], Shiping Yang[4]**

[1]Halicioğlu Data Science Institute, University of California, San Diego
[2]Shenzhen International Graduate School, Tsinghua University
[3]Independent Researcher
[4]School of Computer Science, Beijing University of Posts and Telecommunications
dal034@ucsd.edu, zhang-hy22@mails.tsinghua.edu.cn,
yanranli.summer@gmail.com, yangshipingnlp@gmail.com,

## Abstract

In this work, we tackle the scenario of understanding characters in scripts, which aims to learn the characters' personalities and identities from their utterances. We begin by analyzing several challenges in this scenario, and then propose a multi-level contrastive learning framework to capture characters' global information in a fine-grained manner. To validate the proposed framework, we conduct extensive experiments on three character understanding sub-tasks by comparing with strong pre-trained language models, including SpanBERT, Longformer, BigBird and ChatGPT-3.5. Experimental results demonstrate that our method improves the performances by a considerable margin. Through further in-depth analysis, we show the effectiveness of our method in addressing the challenges and provide more hints on the scenario of character understanding. We will open-source our work in this URL.

## 1 Introduction

As one essential element in stories, character comprehension is a popular research topic in literary, psychological and educational research (McKee, 1997; Currie, 2009; Paris and Paris, 2003; Bower, 1978). To fully understand characters, individuals must empathize with characters based on personal experiences (Gernsbacher et al., 1998), construct profiles according to characters' identities, and inference about characters' future actions (Fiske et al., 1979; Mead, 1990).

According to the data modality and format, character comprehension can be categorized into several classes (Sang et al., 2022a). In this work, we focus on character understanding in scripts (Chen and Choi, 2016; Sang et al., 2022b). Scripts are written text for plays, movies, or broadcasts (Onions et al., 1966). Typically, scripts are often structured with several text fields, including scene description, conversation, transition and summary (Saha, 2021).

| Character | Sheldon | Jennifer |
|---|---|---|
| Story Title | TBBT | The Test |
| Dataset | TVSHOWGUESS | ROCStories |
| Text Length | 528832 | 41 |
| Character's Related Text | Sheldon: " ... we take on Koothrappaliand his dog. Really give ourselves a challenge." | Jennifer has a big exam tomorrow. ... Jennifer felt bittersweet about it. |

Table 1: Comparison between a script from TVSHOWGUESS (Sang et al., 2022b) and a narrative from ROCStories (Mostafazadeh et al., 2016).

Although pre-trained language models (PLMs) have demonstrated their effectiveness in language and vision research fields (Qiu et al., 2020; Min et al., 2023), script-based character understanding is yet a hard task, as shown in our experiments. Here we highlight two challenges. The first one is **text type**. As scripts mainly consist of conversations between different characters, at the core of script-based character understanding is conversation understanding. Especially, scripts often involve multi-party conversations where multiple characters talk and interact with each other in a single scene. Considering other common issues in conversation understanding, it is non-trivial for PLMs to comprehend characters based on fine-grained conversation information (Li and Zhao, 2021; Ma et al., 2022; Li et al., 2022; Tu et al., 2022). The other challenge of applying PLMs to script-based character understanding is **text length**. Table 1 shows a comparison between a script from TVSHOWGUESS (Sang et al., 2022b) and a short story from ROCStories (Mostafazadeh et al., 2016). Typically, scripts are very long with even billion of words (Chen and Choi, 2016), and in turn character information are distributed globally throughout the entire script (Bai et al., 2021; Inoue et al., 2022). However, PLMs are ineffective in capturing such global information due to the sensitiveness of context modeling (Liu et al., 2019; Joshi et al., 2020)

and the limitation of input length (Dai et al., 2019; Beltagy et al., 2020).

To address the aforementioned challenges, we propose a multi-level contrastive learning framework and capture both fine-grained and global information using two devised contrastive losses. For fine-grained character information, we build a **summary-conversation contrastive loss** by comparing character representations from different sources. Specifically, we leverage two text fields in scripts, i.e., summary and conversation, and then extract character representations from the corresponding field. The representations of the same character are then treated as the postive pairs, while those of different characters are negative pairs. To model the global information, we also propose a novel **cross-sample contrastive loss** as inspired by (Bai et al., 2021; Inoue et al., 2022). By aligning the same character's representation in different samples, the model overcomes the input length limitation and learns the global information of each character. To validate the effectiveness of our framework, we benchmark the performances of several PLMs, including SpanBERT, Longformer, BigBird, and ChatGPT-3.5, on three widely-adopted character understanding tasks.

In general, our contributions are as follows:
- We identify two critical challenges for character understanding in scripts and propose a multi-level contrastive learning framework to address them.
- Through extensive experiments, we demonstrate the effectiveness of our method across multiple datasets and downstream tasks.
- With further analysis, we provide some insights into script-based character understanding. All codes will be open-sourced for future research.

## 2 Related Work

### 2.1 Character Understanding

Character understanding has long been the subject of considerable interest and scrutiny. Some early works propose to extract keywords as characters' features from movies (Bamman et al., 2013) and novels (Flekova and Gurevych, 2015). Other works attempt to learn the relationship between characters in both supervised (Massey et al., 2015; Kim and Klinger, 2019) and unsupervised ways (Krishnan and Eisenstein, 2015; Iyyer et al., 2016).

Recently, more challenging tasks in character understanding have emerged. Chen and Choi (2016)

benchmark the character linking and coreference resolution tasks on TV show scripts. Brahman et al. (2021) collect dataset with storybooks and their summaries, and define the character description generation and character identification tasks. Sang et al. (2022b) extend the character guessing task into a multi-character scenario on TV show scripts. Additionally, some works attempt to combine traditional self-supervised learning methods (Mikolov et al., 2013) with language models (Liu et al., 2019) to learn contextual character embeddings and apply them in downstream tasks (Azab et al., 2019; Inoue et al., 2022).

In this work, we focus on character understanding tasks in scripts. While some works benchmark summary-based tasks in narratives (Chen et al., 2022; Brahman et al., 2021), we are the first to leverage script summaries as auxiliary data and learn fine-grained and global character representations in a novel way.

### 2.2 Contrastive Learning

In recent years, contrastive learning is widely used in various NLP tasks (Zhang et al., 2022b), including sentence representation (Gao et al., 2021; Kim et al., 2021), machine translation (Pan et al., 2021; Vamvas and Sennrich, 2021), text generation (Lee et al., 2020; Shu et al., 2021; Zhang et al., 2022a, 2023), and etc. Literatures in multimodal research field adopt contrastive learning for vision-language model training, constructing positive pairs with images and their corresponding captions (Li et al., 2020; Radford et al., 2021; Yang et al., 2022). In our work, we also regard characters in summaries and conversations as two different views of the same target and align them for a better representation.

Moreover, some works aim to construct positive pairs in global manners. Both Qin et al. (2021) and Hogan et al. (2022) conduct document-level contrastive learning in the relation extraction task to align the representation of the same entity and relation. Pan et al. (2021) propose an aligned augmentation method that generates more positive sentence pairs in different languages to improve translation performances in non-English directions. Similarly, Qin et al. (2022) acquire multilingual views of the same utterance from bi-lingual dictionaries. Following this line of research, we propose the cross-sample contrastive learning in addition to the in-sample contrastive loss to learn character

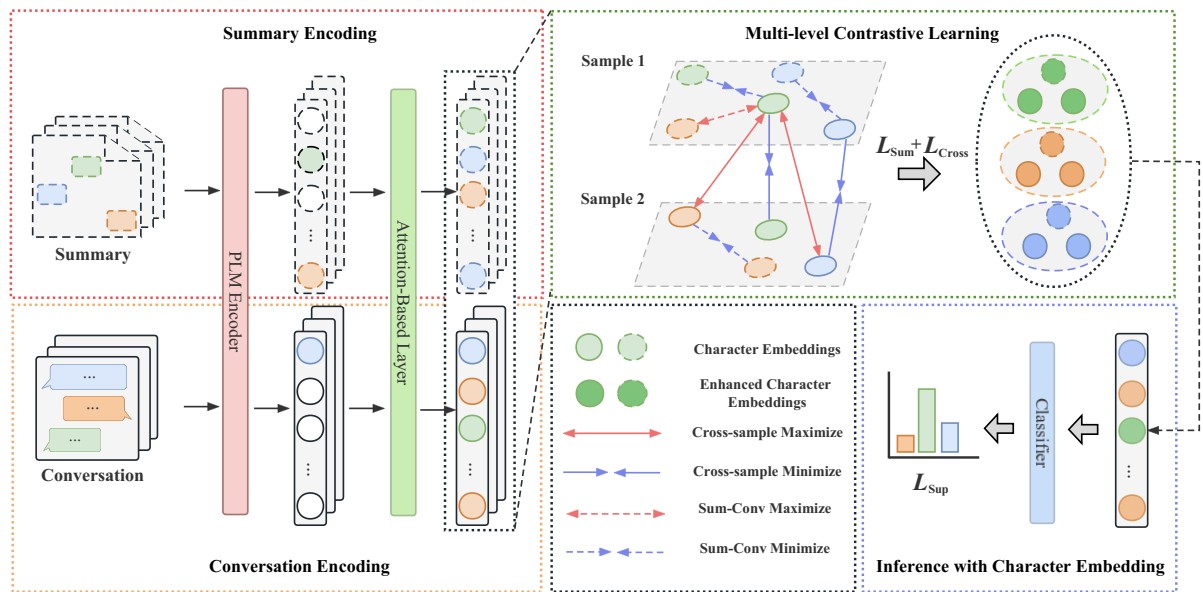

Figure 1: The overview pipeline of our method. Each color represents a character entity or embedding. The conversation and summary encoding parts correspond to Section 4.1 and 4.2 respectively. The multi-level contrastive learning part corresponds to Section 4.3. The inference with character embedding part corresponds to Section 4.4.

representations globally.

## 3 Preliminaries

Generally, character understanding tasks require the model to predict character's information given a segment of text. For script-based character understanding, the provided texts often consist of conversations within scripts. In this work, we also leverage script summaries as an additional source. We provide detailed examples in Appendix A.

In practice, the model first generates character's embeddings $e$ in the representation learning step. Subsequently, a feed-forward network FFN is often adopted as the classifier with the cross-entropy loss:

$$p = Softmax(\text{FFN}(e)) \qquad (1)$$

$$L_{Sup} = -\frac{1}{N} \sum_{i=1}^{N} y_i \log(p) \qquad (2)$$

## 4 Method

Our work presents a multi-level contrastive learning framework for character representation learning. Firstly, we follow a general encoding process to obtain character representations from conversations and summaries. Then, we describe two novel contrastive losses to capture fine-grained and global information at both in-sample and cross-sample levels. Finally, we propose a two-stage training

paradigm that applies different losses in different learning stages. Figure 1 illustrates an overview pipeline of our method.

### 4.1 Character Representation in Conversation

To obtain character representations from the conversation field in the scripts, we first concatenate each utterance (Joshi et al., 2020; Beltagy et al., 2020) and utilize a pre-trained language model PLM[1] to produce the encoding of the whole text $\mathbf{H}$:

$$\mathbf{H} = \text{PLM}(u_1; u_2; , ...; u_T) \qquad (3)$$

Then, the character embeddings $e_1, e_2, ...e_n$ are extracted from the contextual encoding $\mathbf{H}$. After that, we follow previous works (Bai et al., 2021; Sang et al., 2022b) and use an attention-based layer to share the character-level information among each embedding[2]:

$$e_1, ...e_n = \text{Extract}(\mathbf{H}) \qquad (4)$$

$$e_1, ...e_n = \text{Attention}(e_1, ...e_n) \qquad (5)$$

However, the conversations in the scripts are complex and thus the character embeddings solely based on the conversations are often insufficient for fine-grained character understanding.

---

[1]Without loss of generalization, we adopt several PLMs in experiments.

[2]We provide further details in Appendix B

## 4.2 Character Representation in Summary

To supply more information, we leverage scripts' summaries as auxiliary data and apply contrastive learning to capture the character intricacies.

Similar with conversation encoding, given a summary $S$ contains a group of character mentions $\{cm_1^s, cm_2^s, ..., cm_n^s\}$, we also encode the whole summary and extract the character representations:

$$\mathbf{H}_s = \text{PLM}(S) \tag{6}$$

$$e_i^s = t_{start_i} + t_{end_i}, 1 <= i <= n \tag{7}$$

where $t_{start_i}$ and $t_{end_i}$ are the first and last tokens of the $i_{th}$ character mention $cm_i^s$ in the summary.

After that, we follow (Bai et al., 2021) and use a mention-level self-attention (MLSA) layer [3] to gather information for each character embedding:

$$e_1^s, ..., e_n^s = \text{MLSA}(e_1^s, ..., e_n^s) \tag{8}$$

and the last layer's output $e_i^s$ is treated as the character's representation from the summary.

## 4.3 Multi-level Contrastive Learning

To enhance the character representations learned from the conversation and the summary, we develop a novel multi-level contrastive learning to capture both fine-grained and global information.

### 4.3.1 Summary-conversation Contrastive Learning

At the local in-sample level, we develop a summary-conversation contrastive loss to align representations of the same character. This gives the model an additional perspective on character representation and encourages it to find a general space where different representations of the same character are closer. Concretely, the loss function for the summary-conversation contrastive learning is:

$$L_{Sum} = \sum_{i=1}^{P} -log \frac{exp^{\text{sim}(e_{c_i}, e_{c_i}^s)}/\tau}{\sum_{j=1}^{P} exp^{\text{sim}(e_{c_i}, e_{c_j}^s)}/\tau} \tag{9}$$

where $c_i$ denotes the $i_{th}$ character, and $P$ here is the number of characters that appear in both scripts and summaries. Also, $\tau$ is a temperature hyper-parameter, and $\text{sim}(,)$ stands for the similarity function[4]. Note that in samples where conversation and summary contain multiple representations of

character $c_i$, we randomly select one as $e_{c_i}$ and $e_{c_i}^s$, respectively.

By applying the summary-conversation contrastive loss, we are able to learn fine-grained character representations from both summary and conversation texts.

### 4.3.2 Cross-sample Contrastive Learning

In addition to fine-grained information, global-level information is also crucial for character representation learning (Bai et al., 2021; Inoue et al., 2022). To this end, we also propose a cross-sample contrastive learning to align the same character representation in different samples within a batch:

$$L_{Cross} = \sum_{i=1}^{K} -log \frac{exp^{\text{sim}(e_{c_i}^1, e_{c_i}^2)}/\tau}{\sum_{j=1}^{K} exp^{\text{sim}(e_{c_i}^1, e_{c_j}^2)}/\tau} \tag{10}$$

$$SI(e_{c_i}^1) \neq SI(e_{c_i}^2) \tag{11}$$

where $SI(e)$ means the sample index of the character representation $e$[5]. When there are multiple representations of one given character in a batch, we randomly select two from them. For cross-sample learning, we impose a constraint that restricts $e_{c_i}^1$ and $e_{c_i}^2$ to originate from different samples. $K$ is the number of characters appearing in at least two different samples within a batch. To this end, the cross-sample contrastive loss forces the model to utilize global information in a batch and thus obtain a comprehensive understanding of the characters.

## 4.4 Two-stage Training

To fully train the model, we further propose a two-stage training paradigm to apply different losses in different learning stages.

Concretely, in the first stage, we combine the two contrastive losses with the supervised loss together, and post-train the pre-trained language model. The supervised loss serves as a guidance to facilitate the contrastive learning, and stabilize the training at the very beginning. The total loss of the first stage is:

$$L_{Total} = \lambda * L_{Sup} + \alpha * L_{Sum} + \beta * L_{Cross} \tag{12}$$

where $\lambda, \alpha, \beta$ are hyper-parameters of task ratios, and we will analyze their effects in Section 6.3. After the first stage, only the supervised loss is

---

[3]It is a transformer encoder layer with $B$ repeated block. Please refer to Bai et al. (2021) for more details.

[4]Here we use Cosine similarity.

[5]$e$ generally represents any character embedding.

kept to train the model in the second stage. This makes the model concentrate on the downstream supervision signals.

## 5 Experiments Setup

### 5.1 Tasks and Datasets

We evaluate the proposed method on three character understanding tasks, i.e., coreference resolution (Chen and Choi, 2016), character linking (Chen and Choi, 2016), and character guessing (Sang et al., 2022b).

**Coreference Resolution** Given a conversation in scripts that contains multiple utterances and $n$ character mention entity $c_1, c_2, ..., c_n$ within it, the objective of the coreference resolution task is to assemble all mention entities that refer to the same character in a cluster.

**Character Linking** The input of the character linking task is the same as coreference resolution. Unlike coreference resolution, the goal of character linking is to accurately classify each mention entity to the character in a pre-defined character set $Z = \{z_1, z_2, ..., z_m\}$.

**Character Guessing** Distinct from previous tasks, the character guessing task focuses on identifying the speaker for each utterance in scripts. In this task, each utterance within a scene is segmented and fed into the model. The speaker's name preceding each utterance is masked and replaced with a special token. The same speaker within a scene is represented by the same special token. The objective of the character guessing task is to predict the identity of the speaker for each special token.

**Datasets** We choose two TV show datasets to conduct experiments. For coreference resolution and character linking, we use the latest released version of the Character Identification dataset[6]. For character guessing, we adopt the TVSHOWGUESS dataset[7] to conduct experiments. We follow all the training, development, and testing separation provided by the original datasets. The dataset statistics are given in Table 13 in Appendix.

### 5.2 Baseline Models

Following previous works, we adopt several state-of-the-art (SOTA) models in character understanding as baselines and apply the proposed framework on them. For coreference resolution and

character linking, we choose **SpanBERT** (Joshi et al., 2020), a transformer-architecture pre-trained model with the contiguous random span mask strategy in the pre-training stage. We also adopt $\mathbf{C^2}$, which combines coreference resolution and character linking together and achieves the SOTA performance in both two tasks. For character guessing, we use **BigBird** (Zaheer et al., 2020) and **Longformer** (Beltagy et al., 2020), as they are specialized for long-form document input. We follow Sang et al. (2022b) and add a character-specific attentive pooling layer upon the the model encoders and denote them as **BigBird-P** and **Longformer-P**. Notably, we also design a zero-shot and one-shot instruction prompts and evaluate **ChatGPT-3.5** (gpt-3.5-turbo) via its official API[8] as another strong large language model baseline.

### 5.3 Evaluation Metrics

For coreference resolution, we follow the previous works (Zhou and Choi, 2018; Bai et al., 2021) and use B3, CEAF$\phi$4, and BLANC as our evaluation metrics. These three metrics are first proposed by the CoNNL'12 shared task (Pradhan et al., 2012) to measure the clustering performance of the coreference resolution task. For character linking and character guessing, we use Macro and Micro F1 to evaluate the models' classification performances.

### 5.4 Implementation Details

We employ both the base and large sizes of each model, and implement our proposed method on them. For summary-conversation contrastive loss, we use summary corpus collected by Chen et al. (2022). We follow the hyper-parameter settings in the original papers to reproduce each baseline's result. We repeat each experiment 3 times and report the average scores. For ChatGPT prompts and other implementation details, please refer to Appendix C and Appendix D. We will open-source all codes in this work.

## 6 Results and Analysis

### 6.1 Main Results

Table 2 and Table 3 present the automatic evaluation results on the three tasks. Surprisingly, even with specialized instruction and one-shot demonstration, ChatGPT-3.5 performs the worst among all the baselines on each task. This implies that

---

[6]https://github.com/emorynlp/
character-identification
[7]https://github.com/YisiSang/TVSHOWGUESS

[8]https://platform.openai.com/docs/
api-reference/completions/create

| MODEL | Coreference Resolution | | | | | | | | | Character Linking | |
|---|---|---|---|---|---|---|---|---|---|---|---|
| | B3 | | | CEAF$\phi$4 | | | BLANC | | | MICRO | MACRO |
| | PREC. | REC. | F1 | PREC. | REC. | F1 | PREC. | REC. | F1 | | |
| ChatGPT-Zero-Shot | 63.43 | 59.51 | 61.41 | 68.39 | 64.37 | 66.32 | 80.39 | 77.74 | 78.97 | 74.7 | 64.3 |
| ChatGPT-One-Shot | 66.43 | 62.54 | 64.43 | 68.47 | 64.44 | 66.40 | 82.19 | 79.40 | 80.70 | 76.2 | 63.6 |
| SpanBERT-base | 77.40 | 82.67* | 79.94 | 74.69 | 67.93 | 71.15* | 84.80* | 89.96 | 87.20 | 85.0* | 78.4 |
| SpanBERT-base (Ours) | 79.95* | 84.71 | 82.26 | 76.67 | 70.38 | 73.39* | 87.44 | 91.26 | 89.26 | 86.3 | 78.9* |
| SpanBERT-large | 81.92 | 85.56 | 83.69* | 77.85 | 74.74 | 76.25* | 88.61* | 91.91 | 90.20 | 87.2* | 82.8* |
| SpanBERT-large (Ours) | 83.55* | **87.38*** | 85.42* | **79.83** | 76.29 | 78.02* | 89.18* | 93.00 | 91.00 | **88.2*** | **83.7*** |
| C$^2$-base | 80.75 | 84.77* | 82.71* | 76.97 | 71.78 | 74.28 | 82.22* | 91.52 | 89.80* | 85.6 | 80.4* |
| C$^2$-base (Ours) | 83.35 | 85.12* | 84.23 | 76.88* | 74.97 | 75.91 | 90.48 | 91.85* | 91.15 | 86.4 | 81.1 |
| C$^2$-large | 84.98 | 86.92* | 85.94 | 79.63 | 78.16* | 78.89 | 90.87* | 93.05* | 91.93 | 87.6* | 82.5* |
| C$^2$-large (Ours) | **86.42** | 86.44* | **86.24*** | 78.82 | **80.42*** | **79.61** | **91.77*** | **93.13** | **92.45*** | 88.0* | 83.2* |

Table 2: Automatic evaluation results on coreference resolution and character linking. The best results are in bold. We follow previous works to present the results of coreference resolution in a 2-digital decimal and the results of character linking in a 1-digital decimal. * denotes that $p \leq 0.01$ in the statistical significance test.

| Model | MICRO | MACRO |
|---|---|---|
| ChatGPT-Zero-Shot | 48.58 | 42.17 |
| ChatGPT-One-Shot | 51.57 | 44.05 |
| BigBird-P-base | 71.01 | 70.32* |
| BigBird-P-base (Ours) | 72.61 | 73.00 |
| BigBird-P-large | 75.43* | 75.24 |
| BigBird-P-large (Ours) | 77.68* | 76.41 |
| Longformer-P-base | 71.80 | 73.75 |
| Longformer-P-base (Ours) | 73.65* | 74.22 |
| Longformer-P-large | 77.58 | 75.92* |
| Longformer-P-large (Ours) | **78.92*** | **76.52*** |

Table 3: Automatic evaluation results on character guessing. The best results are in bold.

character understanding is still hard and complex to solve for large language models. Among the three tasks, models perform worse on character guessing than on coreference resolution and character linking tasks. In particular, ChatGPT achieves extremely low scores of 44.05 Macro-F1 in character guessing. Since character guessing requires a deeper understanding of each character and more varied narrative comprehension skills (Sang et al., 2022b), this suggests that **the current pre-trained models, especially LLMs, have room for improvement in tasks that require global and in-depth learning for a specific individual**.

Despite the discrepancies in model architecture and size, the proposed method brings significant improvements to each baseline model on almost every metric, except for B3 and CEAF$\phi$4 in C$^2$-large model. These results indicate the effectiveness and compatibility of our method.

## 6.2 Ablation Studies

We also conduct an ablation study to examine the contributions of the two novel contrastive losses, i.e., the cross-sample loss and summary-conversation loss. To implement, we select SpanBERT-base and SpanBERT-large as backbone models and implement model variants by removing one of two contrastive losses in the training phases.

Table 4 presents the results of our ablation study on the coreference resolution and character linking tasks. Compared with the vanilla SpanBERT-base and SpanBERT-large, adding one or two contrastive losses yield better performances. Additionally, we observe that when applied separately, models with the summary-conversation loss work better than models with the cross-sample loss only. More importantly, it is evident that the models trained with both contrastive losses together outperform the models with only one loss, indicating the necessity of our multi-level contrastive framework as well as its effectiveness in addressing the two challenges, i.e., text type and text length.

We also conduct an ablation study on the two-stage learning strategy. Table 5 shows the experiment results on C2-base using character linking and coreference resolution. While the one-stage multi-task training can also improve the baseline model's performance, we found it leads to a sub-optimal result compared with that using our two-stage learning strategy. This observation leads us to the conclusion that supervision-only fine-tuning is also very important in our method, consistently enhancing baseline models' performance. This aligns with the findings of prior research, which advocate for task-specific fine-tuning following multi-task

| MODEL | Coreference Resolution | | | | | | | | | Character Linking | |
|---|---|---|---|---|---|---|---|---|---|---|---|
| | B3 | | | CEAF$\phi$4 | | | BLANC | | | MICRO | MACRO |
| | PREC. | REC. | F1 | PREC. | REC. | F1 | PREC. | REC. | F1 | | |
| SpanBERT-base | 77.40 | 82.67 | 79.94 | 74.69 | 67.93 | 71.15 | 84.80 | 89.96 | 87.20 | 85.0 | 78.4 |
| SpanBERT-base (Ours) | **79.95** | **84.71** | **82.26** | **76.67** | 70.38 | **73.39** | 87.44 | **91.26** | **89.26** | **86.3** | 78.9 |
| w/o cross-sample loss | 79.48 | 83.06 | 81.23 | 74.72 | **71.07** | 72.85 | **87.68** | 90.59 | 89.08 | 86.1 | **80.0** |
| w/o summary-conversation loss | 79.00 | 83.36 | 81.11 | 74.68 | 70.33 | 72.44 | 85.45 | 90.61 | 87.85 | 85.6 | 78.8 |
| SpanBERT-large | 81.92 | 85.56 | 83.69 | 77.85 | 74.74 | 76.25 | 88.61 | 91.91 | 90.20 | 87.2 | 82.8 |
| SpanBERT-large (Ours) | 83.55 | **87.38** | **85.42** | **79.83** | 76.29 | **78.02** | 89.18 | **93.00** | **91.00** | **88.2** | **83.7** |
| w/o cross-sample loss | **83.85** | 86.68 | 85.24 | 79.44 | 76.37 | 77.88 | **90.68** | 92.47 | 90.65 | 87.4 | 83.6 |
| w/o summary-conversation loss | 85.29 | 83.96 | 84.62 | 74.65 | **79.20** | 76.69 | 91.45 | 91.15 | 90.80 | 87.9 | 82.8 |

Table 4: Ablation study results on two contrastive losses. The experiment is conducted using character resolution and character linking.

| | Coreference Resolution | | Character Linking | |
|---|---|---|---|---|
| | B3 | CEAF$\phi$4 | BLANC | MICRO | MACRO |
| C2-base | 82.71 | 74.28 | 89.80 | 85.6 | 80.4 |
| C2-base-OS | 83.58 | 74.28 | 90.63 | 86.1 | 80.8 |
| C2-base-TS | **84.23** | **75.91** | **91.15** | **86.4** | **81.1** |

Table 5: Ablation study results on two-stage learning strategy. -OS and -TS represents the one-stage training and two-stage training respectively. For one-stage training, we remove the second supervised loss-only stage and adopt the multi-task training only.

post-training (Guan et al., 2020; Han et al., 2021).

| $\lambda$ | $\alpha$ | $\beta$ | B3 | CEAF$\phi$4 | BLANC | MICRO | MACRO |
|---|---|---|---|---|---|---|---|
| – | – | – | 83.69 | 76.25 | 90.20 | 87.2 | 82.8 |
| 1.0 | 0.0 | 0.0 | 83.98 | 76.12 | 90.75 | 87.8 | 82.2 |
| 0.0 | 1.0 | 1.0 | 85.04 | 77.72 | 90.77 | 86.4 | 78.6 |
| 1.0 | 1.0 | 1.0 | 85.42 | 78.02 | 91.00 | 88.2 | 83.7 |
| 0.5 | 1.0 | 1.0 | 85.14 | 78.15 | 91.02 | 88.4 | 83.2 |
| 1.0 | 0.5 | 0.5 | 85.23 | 79.00 | 90.96 | 88.1 | 82.0 |

Table 6: Hyper-parameter analysis results on coreference resolution and character linking. For coreference resolution, we report the F1 scores of the B3, CEAF$\phi$4 and BLANC metrics.

### 6.3 Analysis on Hyper-Parameters

The task ratio setting is also an important component of our method. In this section, we investigate their impacts by testing various task ratios in the first training stage. We employ the SpanBERT-large model and perform experiments on the coreference resolution and character linking tasks.

The results of the hyper-parameter analysis are presented in Table 6. As defined in Equation 12, $\lambda$, $\alpha$, and $\beta$ represent the ratios of task-specific supervised loss, summary-conversation loss, and

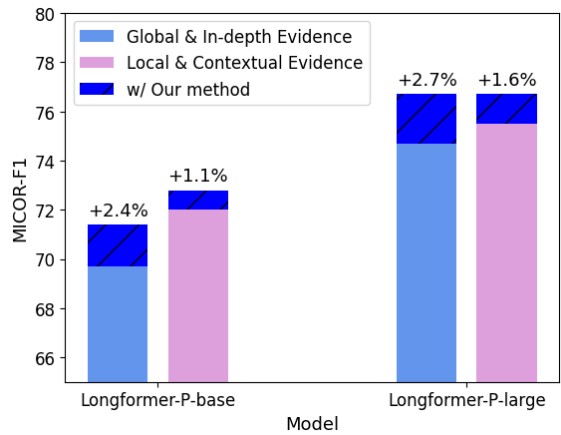

Figure 2: Evidence type analysis result.

cross-sample loss, respectively. Accordingly, the first block (Row 1) presents the vanilla SpanBERT-large performance w/o our framework, and the second block (Row 2 and Row 3) shows the model variants with only supervision loss or contrastive losses. Comparing the first and second block we can see, there is no obvious improvement when only keeping the supervised loss, a.k.a $\lambda = 1.0, \alpha = 0.0, \beta = 0.0$ in the first stage. Moreover, when $\lambda$ is set to 0.0, the model trained without supervised loss also exhibits inferior performances, e.g., there is a notable decrease in Macro F1 (from 82.8 to 78.6). This finding supports our hypothesis that **the task-specific supervision signal plays a crucial role in guiding the two contrastive learning**. When examining the last block (Row 4-6), we observe that the models w/ our framework under different task ratios consistently surpasses the others (except only one MARCO metric). This further demonstrates the robustness of our method on the task ratio hyper-parameter.

## 6.4 Resource Availability Analysis

The proposed summary-conversation contrastive learning relies on well-organized script datasets that include a summary of each scene. This prerequisite could potentially limit the applicability of our approach to datasets in other languages or domains. To address this constraint, we conduct an experiment in which we replaced the manually collected summary dataset with an automatically generated one, produced by ChatGPT. As depicted in Table 7, our results indicate that when using the auto-generated corpus in summary-conversation contrastive learning, a significant improvement is still observed when compared to the vanilla baseline. This discovery further validates the adaptability of our method, irrespective of whether golden or generated summaries are used.

|  | Coreference Resolution | | | Character Linking | |
|---|---|---|---|---|---|
|  | B3 | CEAF$\phi$4 | BLANC | MICRO | MACRO |
| C2-base | 82.71 | 74.28 | 89.80 | 85.6 | 80.4 |
| C2-base-LLM | 84.14 | **76.06** | 90.89 | 86.1 | 80.9 |
| C2-base-G | **84.23** | 75.91 | **91.15** | **86.4** | **81.1** |

Table 7: Experiment results with automatically generated summarization. -LLM and -G denote the model trained on summaries generated by ChatGPT and those trained using the dataset provided by (Chen et al., 2022).

## 6.5 Breakdown to Evidence Type

To better understand when and how our method works on each sample, we conduct an evidence type analysis on the character guessing task based on the fine-grained annotation provided by Sang et al. (2022b). To remedy the scarcity issue in the original annotations, we merge the fine-grained annotation categories into two broader categories: *Global & In-depth Evidence* and *Local & Textual Evidence*. More details on evidence type merging is described in Appendix E.

The results of evidence type analysis are presented in Figure 2. Note that our framework works better when Local & Textual evidence is required for character guessing than Global & In-depth evidence. This finding aligns with our intuition that Global & In-depth evidence is more challenging for the model to comprehend. It is also worth noting that our framework yields larger increases for samples requiring Global & In-depth evidences (2.4% and 2.7% for the base and large size models respectively), as compared to those requiring Local

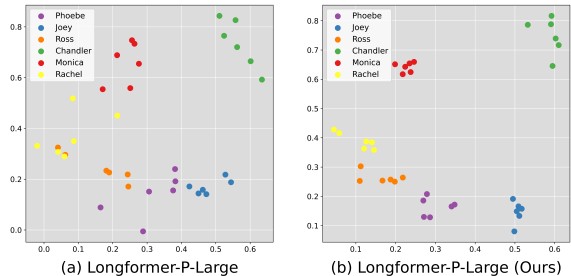

Figure 3: Character embedding visualization result.

& Textual evidence (1.1% and 1.6% for the base and large models respectively). Based on these results, we safely conclude that **our framework is effective in facilitating character information modeling, especially for global information**.

## 6.6 Visualization

The core of our method is to learn fine-grained and global character representations. To this end, we also visualize the learned character embeddings in the character guessing task. Specifically, we use character embeddings in the test set of the "FRIENDS" (a subset of TVSHOWGUESS dataset) and randomly choose 6 embeddings for each character from different samples.

Figure 3 shows the visualization results using T-SNE (Van der Maaten and Hinton, 2008). We compare the character embeddings generated by Longformer-P-Large w/ and w/o our framework. One thing to note is that without our framework, some character embeddings of Ross overlap with those of Rachel. This is because that in the TV show "FRIENDS", Ross and Rachel are partners and together appearing and engaging in many scenes. In contrast, this overlapping phenomenon is greatly mitigated. Overall speaking, our framework encourages the embeddings belonging to the same character exhibit a more compact clustering pattern. This finding provides a new perspective to understand the effectiveness of our proposed method in character comprehension tasks.

## 6.7 Case Study

We also choose a challenging sample from "The Big Bang Theory" subset of TVSHOWGUESS in the character guessing task, and analyze the predictions from Longformer-P-Large w/o and w/ our method, as well as that from ChatGPT.

As shown in Table 8, all the predictions from ChatGPT are wrong, indicating ChatGPT lacks a fine-grained understanding of each character. Be-

sides, the only difference between the vanilla model w/ and w/o our framework is whether the speaker P1 is predicted correctly or not. In this case, predicting P1 is particularly challenging, as few utterances are spoken by this character. Hence, it is a must for the models to guess P1's identity using other details in the scene. By understanding the relationships between P1 and other characters, our method is able to correctly predict that P1 is Sheldon's partner, Amy. This demonstrates that **our method benefits the fine-grained understanding on character relationships in script-based character understanding**, e.g., character guessing tasks.

---

P0 : Hey, sorry about that
P1 : No, we're sorry. We never should have been comparing relationships in the first place.
P2 : Why? We won. You know, I say, next, we take on Koothrappali and his dog. Really give ourselves a challenge.
P3 : I just want to say one more thing about this. Just because Penny and I are very different people does not mean that we're a bad couple.
P2 : The answer is one simple test away. Hmm? You know, it's like when I thought there was a possum in my closet. Did I sit around wondering? No, I sent Leonard in with a pointy stick and a bag.
P3 : I killed his Chewbacca slippers.
P0 : Let's just take the test.
P3 : No, no, no, I don't want to.
P0 : Oh, well, 'cause you know we're gonna do bad.
P3 : Because it doesn't matter. I don't care if we're a ten or a two.
P2 : Or a one. A one is possible.
P3 : Marriage is scary. You're scared, I'm scared. But it doesn't make me not want to do it. It, it just makes me want to hold your hand and do it with you.
P0 : Leonard.
P1 : It makes me so happy if you said things like that.
P2 : We got an eight-point-two. Trust me, you're happy.

---

ChatGPT:*P0: Leonard, P1: Sheldon, P2: Penny, P3:Howard*

---

Vanilla: *P0: Penny, P1: Howard, P2: Sheldon, P3:Leonard*

---

Ours: *P0: Penny, P1: Amy, P2: Sheldon, P3:Leonard*

---

Golden: *P0: Penny, P1: Amy, P2: Sheldon, P3:Leonard*

---

Table 8: An example chosen from "The Big Bang Theory" in the character guessing task. We analyze the predictions made by ChatGPT (one-shot), Longformer-P-Large (vanilla and with our framework).

## 7   Discussion about LLMs on Character Understanding

In this section, we go deeper to discuss the unsatisfied performance when adopting the ICL of LLMs to perform character understanding tasks. One possible reason for this is the script-based character understanding we focus on requires the model to learn the character information globally. For example, in character guessing, anonymous speakers sometimes need to be identified with some global evidence, like linguistic style and the character's relationship with others. These subtle cues are usually not included in the current sample and thus require the model to learn them globally from other samples (Sang et al., 2022b). However, due to the fine-tuned unavailability of ICL, LLMs can only utilize local information from the current sample and limited demonstrations to make inferences. We believe this is the reason that LLMs don't perform well in our script-based character understanding scenario. Additionally, we notice ICL also falls short in some other tasks that involve learning a domain-specific entity or individual across multiple samples, like knowledge graph completion (Yao et al., 2023). This shortcoming in the global learning scenario, which is similar to hallucination (Yang et al., 2023) and the reverse problem (Berglund et al., 2023), can limit LLMs' application in many downstream tasks.

It appears that augmenting the number of demonstrations in the prompt could be a potential strategy for enhancing the capabilities of LLMs in these global learning tasks. Nonetheless, it's essential to note that incorporating an excessive number of relevant samples as demonstrations faces practical challenges, primarily due to constraints related to input length and efficiency considerations. In the future, more efforts are needed to explore optimal ways of harnessing the ICL method of LLMs in such global learning scenarios.

## 8   Conclusions

In this work, we focus on addressing two key challenges, text length and text type in script-based character understanding. To overcome these challenges, we propose a novel multi-level contrastive framework that exploits in-sample and cross-sample features. The experimental results on three tasks show that our method is effective and compatible with several SOTA models. We also conduct in-depth analysis to examine our method detailedly and provide several hints in the character understanding tasks.

In the future, we plan to apply contrastive learning to other long-form document understanding tasks, such as long document matching (Jiang et al., 2019) and fiction understanding (Yu et al., 2023).

## 9 Limitations

Our framework depends on pre-trained large languages (PLMs) to encode conversations and summaries, and requires gradient information to tune the PLMs' parameters. This makes it challenging to apply our approach to language models with gigantic sizes. In this work, we demonstrate the generalization of our method in the experimental section at the base and large size, as well as the incapability of ChatGPT-3.5 on character understanding tasks. Nevertheless, it remains unclear how well our framework will fit to 3B+ encoder-decoder PLMs or decoder-only LLMs. As our experiments suggest, there is still room for improvement in character understanding tasks.

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

## A   Example of Script-based Character Understanding Task

| | |
|---|---|
| Input | P0: Hey, sorry about that P1: No, we're sorry. We never should have been comparing relationships in the first place. P2: Why? We won. You know, I say, next, we take on Koothrappali and his dog. Really give ourselves a challenge. P3: I just want to say one more thing about this. Just because Penny and I are very different people does not mean that we're a bad couple. P2: The answer is one simple test away. Hmm? You know, it's like when I thought there was a possum in my closet. Did I sit around wondering? No, I sent Leonard in with a pointy stick and a bag. P3: I killed his Chewbacca slippers. P0: Let's just take the test. P3: No, no, no, I don't want to. P0: Oh, well, 'cause you know we're gonna do bad. P3: Because it doesn't matter. I don't care if we're a ten or a two. P2: Or a one. A one is possible. P3: Marriage is scary. You're scared, I'm scared. But it doesn't make me not want to do it. It, it just makes me want to hold your hand and do it with you. P0: Leonard. P1: It makes me so happy if you said things like that. P2: We got an eight-point-two. Trust me, you're happy. |
| Label | P0: Penny, P1: Amy, P2: Sheldon, P3: Leonard |

Table 9: A example from character guessing task.

| | |
|---|---|
| Input | Ross: I told mom and dad last night, they seemed to take it pretty well. Monica: Oh really, so that hysterical phone call I got from a woman at sobbing 3:00 A.M., "I'll never have grandchildren, I'll never have grandchildren." was what? A wrong number? Ross: Sorry. Joey: Alright Ross, look. You're feeling a lot of pain right now. You're angry. You're hurting. Can I tell you what the answer is? |
| Label-CR | Ross: I told mom and dad last night, they seemed to take it pretty well. Monica: Oh really, so that hysterical phone call I got from a woman at sobbing 3:00 A.M., "I'll never have grandchildren, I'll never have grandchildren." was what? A wrong number? Ross: Sorry. Joey: Alright Ross, look. You're feeling a lot of pain right now. You're angry. You're hurting. Can I tell you what the answer is? |
| Label-CL | I: Ross Geller, mom: Judy Geller, dad: Jack Geller, I: Monica Geller, woman: Judy Geller, I: Monica Geller, I: Monica Geller, Ross: Ross Geller, You: Ross Geller, You: Ross Geller, You: Ross Geller, I: Joey Tribbiani, you: Ross Geller |

Table 10: A example from coreference resolution and character linking tasks. For coreference resolution, the goal of the task is to cluster the coreferences that refer to the same character in one group (we use the same color to represent).

## B   Details of Character Embedding Generation

Here we give detailed formulations of our character embedding extraction and character-level attention process.

**Coreference Resolution & Character Linking** Given the context encoding $\mathbf{H}$, we follow (Bai et al., 2021) to initialize the mention-level character embedding:

$$e_i = t_{start_i} + t_{end_i} + e_{speaker_i} \tag{13}$$

where $t_{start_i}$ and $t_{end_i}$ are the contextualized representation of the beginning and the end tokens of

mention $i$, and the $e_{speaker_i}$ is the speaker embedding for the current speaker of the utterance where the $i_{th}$ mention belong to. The speaker embeddings are randomly initialized before training.

**Character Guessing** We follow (Sang et al., 2022b) to extract speaker-level character embedding from the context encoding $\mathbf{H}$:

$$A = \text{Attention}(\mathbf{H}) \tag{14}$$

$$a_i = \text{Softmax}(A \odot M_i) \tag{15}$$

$$e_i = \mathbf{H}^T a_i \tag{16}$$

where Attention($\cdot$) is a one-layer feedforward network to compute the token-level attention weight. $M_i$ is a token-level mask such that $M_i[j] = 1$ if the $j_{th}$ word belongs to an utterance of the $i_{th}$ anonymous speaker and $Mx[j] = 0$ otherwise. $a_i$ is the token weight used to pool the hidden states to summarize a character representation.

After obtaining character embedding, we adopt the MLSA layer we mentioned in Section 4.2 to gather information for each character embedding:

$$e_1, ..., e_n = \text{MLSA}(e_1, ..., e_n) \tag{17}$$

## C   Prompts for ChatGPT

For character linking, as Table 15 shows, we provide the original scripts' content for ChatGPT, followed by the position of the mention to be inferenced and all the optional characters. For coreference resolution, we tried several different prompts to ask ChatGPT to do clustering. However, there is always omitting of mentions[9] in the models' output which leads to very poor performance on coreference resolution. So we just use the model's output on character linking as the clustering results of each character and calculate the corresponding metrics for coreference resolution.

For character guessing, we provide the show name of the scripts and optional characters to the model. We concatenate them in front of the script's content and input the prompt to ChatGPT to do inference as Table 16 shows. For zero-shot, we try asking ChatGPT to guess one character in each request (represent as Prompt-Character) and guess all characters in each request (represent as Prompt-Sample) and find the latter performs better as Table 11 shows. One possible reason for that is the model would attend to more information from other characters' utterances given Prompt-Sample, which is exactly the key to perform well in character guessing. For one-shot, we only adopt the Prompt-Sample due to its superior performance in the zero-shot setting.

For the one-shot setting, we additionally provide the model a sample together with its label as a demonstration. For both tasks, after getting the output of the model, we also use the RE module provided by Python to map the raw text to the most similar label.

| | MICRO | MACRO |
|---|---|---|
| Prompt-Character | 43.77 | 37.87 |
| Prompt-Sample | **48.58** | **42.17** |

Table 11: Comparison of the two prompt methods in character guessing with zero-shot.

## D   Detailed Training Settings

For every available positive pair [10] of the same character, we randomly choose one to conduct contrastive learning. We set the $\lambda$, $\alpha$ and $\beta$ to 1.0, 0.5, 0.5 respectively in the first stage of training for all three tasks. We test SpanBERT and C$^2$ in coreference resolution and character linking and Longformer-P and BigBird-P in character guessing. We use ChatGPT in all three tasks. Table 12 gives parameter settings of the learning rate, batch size, and training epochs in the two stages of learning. We use Pytorch-1.8.1

---

[9]For example, there are 20 mentions to be clustered in a sample but the model's output just contains the clustering result of 15 of them.

[10]Excluding the situation that a character show in the dialogue but not in the summary, and vice versa.

deep learning framework and Transformers-4.1.2 library for our experiment. We train our models on a single A40 GPU. It takes about 3 hours to train a SpanBert-base/ C$^2$-base model and 20 hours to train a Longformer-P-base/ BigBird-P-base model. Large-version model training takes twice the time. Table 13 shows the detailed statistics of the datasets we use. We also give detailed information about models in the base and large size we use in the experiment in Table 14.

| Dataset | First Stage | | | Second Stage | | |
|---|---|---|---|---|---|---|
| | LR | Batch Size | Epoch | LR | Batch Size | Epoch |
| Coreference Resolution & Character Linking | 1e-5 | - | 30 | 2e-5 | - | 100 |
| Character Guessing | 4e-6 | 4 | 20 | 2e-5 | 2 | 40 |

Table 12: Other hyper-parameters settings in our experiments. Note that for coreference resolution and character linking, we follow the previous works (Chen and Choi, 2016; Bai et al., 2021) and incorporate every sample inside a scene in a batch.

| Task | Train | Validation | Test | Total |
|---|---|---|---|---|
| Character Linking & Coreference Resolution | 987 | 122 | 192 | 1301 |
| Character Guessing | 10071 | 819 | 823 | 11713 |

Table 13: Detailed information about the datasets for each task.

| Size | Parameter Size | Number of Transformer Layer | Number of Attention Head | Hidden Size |
|---|---|---|---|---|
| Base | 125M | 12 | 12 | 768 |
| Large | 354M | 24 | 16 | 1024 |

Table 14: Detailed information about models in the base and large size we use in the experiment.

| | |
|---|---|
| Scripts | Monica Geller: Tell him. |
| | Rachel Green: No. |
| | Phoebe Buffay: Tell him, tell him. |
| | Monica Geller: Just...please tell him. |
| | Rachel Green: Shut up! |
| | Chandler Bing: Tell me what? |
| | Monica Geller: Look at you, you won't even look at him . |
| | Chandler Bing: Oh, come on tell me. I could use another reason why women won't look at me. |
| | Rachel Green: All right, all right, all right. Last night, I had a dream that, uh, you and I, were... |
| | Phoebe Buffay: Doing it on this table . |
| | Chandler Bing: Wow ! |
| | Joey Tribbiani: Exellent dream score . |
| | Ross Geller: Why, why, why would you dream that? |
| | Chandler Bing: More importantly, was I any good? |
| | Rachel Green: Well, you were pretty damn good. |
| | Chandler Bing: Interesting, cause in my dreams, I'm allways surprisingly inadequate. |
| | Rachel Green: Well, last night you seemed to know your way around the table. |
| | Ross Geller: I love it, when we share. Chandler Bing: You're okay there? |
| | Ross Geller: I can't believe you two had sex in her dream. |
| | Chandler Bing: I'm sorry, it was a one-time - thing. I was very drunk and i was somebody else's subconscious . |
| Prompt | Here is an example: |
| | <Demonstration> |
| | Following this example and read the following conversation: |
| | <Scripts> |
| | The NO.1 "him" in the utterance "Tell him ." refer to which character? |
| | you should choose answer from Ross Geller, Rachel Green, Chandler Bing, |
| | Monica Geller, Joey Tribbiani, Phoebe Buffay, Emily, Richard Burke, Carol Willick, |
| | Ben Geller, Peter Becker, Judy Geller, Barry Farber, Jack Geller, Kate Miller, |
| | #OTHER#, #GENERAL# |

Table 15: Original scripts and the prompt we input to ChatGPT in coreference resolution and character linking. We replace the original scripts in the prompt with a special token <Scripts> to save space. We use red color to represent the additional prompt and demonstration adopted in one-shot settings.

| | |
|---|---|
| Scripts | P0 : hey, frasier, i'm glad i caught you. did you just get home?
P1 : no, i've been here a while. can't bring myself to go in. not with her in there.
P1 : she's getting better.
P0 : look, i did you a favor. my lawyer drew up this document, it releases you from all liability, if you just get ann to sign it.
P1 : oh, roz, there's no way i'm going to get her to sign this. but i have a better plan: i've just booked passage for her and her mother on a two week cruise to alaska. that way i'll get her out of my home, but we'll still feel like we're friends.
P0 : hmm, not a bad idea. good luck with that.
P1 : thank you. roz, i've been meaning to ask you: how did you ever become friends with ann? i mean, she's really not your type, is she?
P0 : oh, we're not really friends. i rear ended her in 1989. P1 : oh, the tickets arrived. P1 : well, i'd hoped she would be. P1 : you told her i was taking you?
P1 : ann...
P1 : hold that thought, while bunny goes and pours himself a big ol' glass o' wine. P1 : caroline.
P1 : uh...
P1 : uh...just a neighbor.
P1 : [pained] no. since we made our plans, caroline, i've met someone else.
P1 : just go!
P1 : [darkly] i don't know what i'm gonna do with you either.
P1 : oh, it's nothing. just some work stuff.
P1 : no.
P1 : no!
P1 : no, never!
P1 : i realize that you're angry now, bunny...
P1 : all right, fine! go ahead and sue! i am fed up with this charade! this was an accident! i have cared for you, i have waited on you, i have pumiced your heels and set your hair! well, if that's not enough for you, so be it! i don't care anymore , i will not beg! you can take me to the cleaners but you cannot take my dignity! P1 : oh dear god, please, no! pleas, no, no, please! please, please don't sue me! my...things, my beautiful, beautiful things. i love them so...
P1 : [weeping] no.
P1 : you will?
P1 : thank you, ann. i'm sorry it had to come down to all this, this legal business. if it were up to me, i would tear up this piece of paper and forget everything that's happened here. P1 : and, uh, here.
P1 : and...here. |
| Prompt-Character | Here is an example:
<Demonstration>
Following this example and tell me P0 below are which character from TV show "Frasier",
please choose from frasier, roz, niles, martin, daphne and bob:
<Scripts>
Tell me P1 below are which character from TV show "Frasier",
please choose from frasier, roz, niles, martin, daphne and bob:
<Scripts> |
| Prompt-Sample | Here is an example:
<Demonstration>
Following this example and tell me P0, P1 below are which character from TV show "Frasier",
please choose from frasier, roz, niles, martin, daphne and bob:
<Scripts> |

Table 16: Original scripts and the prompt we input to ChatGPT in character guessing. We replace the original scripts in the prompt with a special token <Scripts> to save space. We use red color to represent the additional prompt and demonstration adopted in one-shot settings.

# E   Evidence Type Merging

In Sang et al. (2022b), the annotations divide the evidence types of guessing characters into extremely detailed categories, and several psychologists are asked to assign a category of evidence to each character. In specific, there are 9 types of evidence totally for guessing character identification according to Sang et al. (2022b). They are attribute, relation, status, background, exclusion, mention, linguistic, memory, and personality.

However, one drawback of the subdivided categories is the scarcity of a certain categories. To address the high variance issue caused by the scarcity, we merge the fine-grained annotations into broader ones. Based on the definition of them in the original paper[11], we split them into 2 big categories as Table 17 shows. Here Global & In-depth evidence means cases in which the character can only be predicted according to his/her global information (like one's relationship with others) or some subtle clues (like one's Linguistic style). Local & Textual evidence means cases in which a character can be easily predicted only using the content in the local sample (like background information appearing in other characters' utterances) or something very direct (like calling one character's name directly). Note that we abandon the evidence exclusion because it is more like a guessing technique rather than evidence.

| Merged Category | Original Category |
|---|---|
| Global & In-depth evidence | Attribute |
| | Relation |
| | Status |
| | Linguistics |
| | Memory |
| | Personality |
| Local & Textual evidence | Background |
| | Mention |

Table 17: Merged result of the subdivided evidence type.

---

[11] Please refer to the original paper for more details.