# OpenReview forum: "Multi-level Contrastive Learning for Script-based Character Understanding"
_EMNLP/2023/Conference — EMNLP 2023 Main_

### Official Review · Reviewer_metd · 2023-08-03

**Soundness:** 3

**Excitement:**

3: Ambivalent: It has merits (e.g., it reports state-of-the-art results, the idea is nice), but there are key weaknesses (e.g., it describes incremental work), and it can significantly benefit from another round of revision. However, I won't object to accepting it if my co-reviewers champion it.

**Paper Topic And Main Contributions:**

This paper proposes a framework for character understanding in scripts which consists of two different constrastive losses and a two-stage training strategy. The two constrative losses are summary-conversation loss and cross-sample contrastive loss, respectively, with the purposes to capture fine-grained and global character information at in-sample and cross-sample levels. Experiments on three different tasks (i.e. coreference resolution, character linking and character guessing) show significant improvements compared to baselines and ChatGPT 3.5 model.

The main contribution of this paper is the two contrastive losses, where we can see their effectiveness in the ablation study. However, as claimed in the limitation section, the summary-conversation loss is limited by the resources. Although authors said summaries could be obtained by text generation, the quality is a concerned issue which might propagate the errors to the task of character understanding.

**Questions For The Authors:**

1. In pracitcal scripts or stories, there are many entities/characters which have alias, e.g. a person could be called Mr. X besides his full name. Do you consider the alias alignment issue (aligning the alias/other mentions to an entity/character) in your work?
2. Why did the summary-conversation loss work better than the cross-sample loss based on the results in Table 4? Please explain the potential resons.
3. In table 4, we can see that without cross-sample loss some results become better than that with it, what are the reasons?
4. How do we know the effectiveness/impact of the two-stage training? Maybe authors can explain something about this in the ablation study.

**Reasons To Accept:**

The main contribution of this paper is the two contrastive losses, where we can see their effectiveness in the ablation study. The two-stage training strategy could be a contribution as well.

**Reasons To Reject:**

Contrastive loss is a straightforward way to enhance the character representation, so the overall novelty of this paper is limited. This work is limited by the resources such as the summary so it is not easy to be generalised to other languages or domains. Using automatically generated texts as summaries is not convincing.

**Reproducibility:**

4: Could mostly reproduce the results, but there may be some variation because of sample variance or minor variations in their interpretation of the protocol or method.

**Reviewer Confidence:**

4: Quite sure. I tried to check the important points carefully. It's unlikely, though conceivable, that I missed something that should affect my ratings.

**Typos Grammar Style And Presentation Improvements:**

To name a few:
1. Line 201: please don't use "get" in the paper because it is a spoken/informal word.
2. Table 2 didn't correctly show the best results in bold, e.g. the best one is 79.83 in Column 4, 88.2 in Column 10 etc.
3. Line 429: separate 'effectivenessin' into two words.

---

> ### Author Rebuttal · Authors · 2023-08-29
>
> We sincerely thank you for dedicating your time and effort to the evaluation of our work. Here we address some rejection reasons and questions you raised with additional experiment results and analysis.
>
> __Our contribution__
> As for the rejection reason, we acknowledge that adopting the contrastive learning method in character understanding and representation learning is quite straightforward. However, we are the first to propose this method in script-based character understanding and evaluate its effectiveness comprehensively. We believe our work will stimulate more focus on adopting contrastive methods to learn character representation.
>
> __Our method with automatically generated summary__
> Regarding your concern about the summary availability, large language models these days demonstrate comparable performance to humans in summarization, according to some recent works[1][2]. To validate our method's compatability with automatically generated summaries, we prompt ChatGPT to automatically generate summary for coreference resolution and character linking. Due to time reasons, we conduct experiment on C2-base and provide the experiment results in the table below. It's significant to highlight that the model trained on automatically generated summaries achieves comparable performance with that trained in human-written summarization. This outcome underscores the potential efficacy of integrating automatically generated summaries into our methodology.
>
> |          | B3        | CEAFφ4    | BLANC     | MICRO    | MACRO    |
> |----------|-----------|-----------|-----------|----------|----------|
> | Baseline | 82.71     | 74.28     | 89.80     | 85.6     | 80.4     |
> | ChatGPT  | 84.14     | **76.06** | 90.89     | 86.1     | 80.9     |
> | Human    | **84.23** | 75.91     | **91.15** | **86.4** | **81.1** |
>
> __Charcter entity extraction__
> Regarding the first question, in our preliminary experiment, we tried using some NER-based methods to extract coreference in summaries and align them to specific characters based on certain rules. However, we found this method would cause quite a lot of errors and influence the subsequent contrastive learning stage. Also, we find this method can’t significantly improve the number of distinct characters extracted in each summary. This is probably due to the fact that each character’s first appearance in the summary is always with his/her name, but not alias. Therefore, we only use character names to extract character representation from summaries.
>
> __Explanation of Table 4__
> To answer the second question, one possible reason for summary-conversation contrastive learning’s better performance is the introduction of additional and different modal information. In our summary-conversation contrastive learning, we adopt summary as auxiliary information to facilitate PLM's understanding of the complex conversation. According to the previous works[3][4], PLM can benefit a lot from such multi-modal information. Compared with it, cross-sample conversation learning only utilizes information within a singular domain (conversation) in a global way.
>
> As for the third question, it is noticeable that although the model without cross-sample contrastive learning performs better in some metrics, our method with the two contrastive losses together achieves the overall best performance considering all the metrics.
>
> __Validation of our two-stage learning__
> Concerning the last question, we tried to train the model using only the first stage with three losses (one supervised and two contrastive losses) together in our preliminary experiment. We found such one-stage learning would lead to a sub-optimal performance and when we add the second stage with solely-supervised training, the performance on the specific task can be further improved. This led us to design a two-stage training and we added some intuitive explanations of it in Section 4.4, lines 296-299. Below we provide the experiment results of baseline/one-stage/two-stage training in coreference resolution and character linking. The model used here is C2-base.
>
> |                  | B3        | CEAFφ4    | BLANC     | MICRO    | MACRO    |
> |------------------|-----------|-----------|-----------|----------|----------|
> | Baseline         | 82.71     | 74.28     | 89.80     | 85.6     | 80.4     |
> | One-stage        | 83.58     | 74.28     | 90.63     | 86.1     | 80.8     |
> | two-stage (Ours) | **84.23** | **75.91** | **91.15** | **86.4** | **81.1** |
>
> We appreciate other suggestions and corrections and kindly request your reconsideration of our paper in light of the aforementioned points.
>
> [1] Gao, Mingqi, et al. "Human-like summarization evaluation with chatgpt." arXiv preprint arXiv:2304.02554 (2023).
>
> [2] Zhang, Tianyi, et al. "Benchmarking large language models for news summarization." arXiv preprint arXiv:2301.13848 (2023).
>
> [3] Radford, Alec, et al. "Learning transferable visual models from natural language supervision." International conference on machine learning. PMLR, 2021.
>
> [4] Zhang, Hengyuan, et al. "Fine-grained Contrastive Learning for Definition Generation." Proceedings of the 2nd Conference of the Asia-Pacific Chapter of the Association for Computational Linguistics and the 12th International Joint Conference on Natural Language Processing. 2022.

---

### Official Review · Reviewer_Cuxc · 2023-08-04

**Typos Grammar Style And Presentation Improvements:** L175
**Soundness:** 4

**Excitement:**

3: Ambivalent: It has merits (e.g., it reports state-of-the-art results, the idea is nice), but there are key weaknesses (e.g., it describes incremental work), and it can significantly benefit from another round of revision. However, I won't object to accepting it if my co-reviewers champion it.

**Missing References:**

N/A

**Paper Topic And Main Contributions:**

The paper proposes a multi-level contrastive learning framework for script-based character understanding. In the in-sample level, they utilize summary of the script and build a summary-conversation contrastive loss. They also provide a cross-sample contrastive loss to capture global feature. Extensive experiments show its effectiveness.

**Questions For The Authors:**

1. What is meant by "fine-grained" representation in L259? Do summaries and conversations convey "fine-grained" information? Summaries can also encompass general and global aspects, especially if "fine-grained" refers to local details.
2. When compared to the baseline model, what are the additional computational costs associated with the two-stage training framework?

**Reasons To Accept:**

1. The motivation behind the paper is clear and justified. The authors acknowledge the significance of different types of text and the importance of long-context descriptions in understanding a figure. The experimental design aligns well with these two aspects.
2. The experiments conducted in the paper are extensive and demonstrate the effectiveness of the proposed method.
3. The structure and organization of the paper are well-designed, resulting in a coherent and easily comprehensible flow of information.

**Reasons To Reject:**

1. While it is acknowledged that text type and longer context are important factors in understanding characters, the connection between these two aspects and contrastive learning seems unclear and somewhat forced in this paper. For text type, the authors choose to focus on summary and conversation, without explicitly justifying why these sources were chosen over others. Additionally, while a summary provides a condensed description of a character, conversations can be random and distributed, which may not always align with the character's summary. This raises concerns about the validity of contrastive learning for positive pairs. Regarding global context, it remains unclear why similar utterance segments from the same person need to be considered positive pairs within the framework. For example, totally different talks may come from the same person.
2. The implementation of cross-sample contrastive learning in this paper appears to be a conventional approach within the field of contrastive learning, lacking novel elements or significant advancements.

**Reproducibility:**

4: Could mostly reproduce the results, but there may be some variation because of sample variance or minor variations in their interpretation of the protocol or method.

**Reviewer Confidence:**

4: Quite sure. I tried to check the important points carefully. It's unlikely, though conceivable, that I missed something that should affect my ratings.

---

> ### Author Rebuttal · Authors · 2023-08-29
>
> Thank you for your detailed evaluation of our work. We appreciate the time and effort you've dedicated to reviewing our paper. Here we would like to clarify our work by explaining the rationale behind our method in more detail.
>
> __Connection between the two challenges and our method__
> Regarding the first issue, we first analyze two challenges of script-based character understanding in Section 1, spanning lines 042-072. We also explain the relationship between the two contrastive losses and two challenges in Section 1, lines 073-096.
>
> Specifically, the first challenge in character understanding is complex text type. To address it, we utilize summary because its straightforward and highly generalized narrative style can be a great supplement to the complex conversation. We use the summary as an auxiliary corpus to prompt the model to better capture fine-grained information (character features) from the conversation.
>
> The second challenge in character understanding we analyze is the script length. We design a cross-sample contrastive learning to address this challenge. It aligns the same character's representation from different samples thus prompting PLM to comprehend character globally.
>
> __Positive pair construction issue__
> As for your concern about the positive pair construction in the two contrastive losses, we recognize the need for further explanation and validation. Firstly, we need to emphasize that no method is perfect, and the construction of positive samples for contrastive learning will definitely introduce some noise. However, we believe the promising experiment results have proven the rationality of our positive sample construction.
>
> Beyond that, samples of each character understanding task are built at the scene level, which means character representation from the conversation is learned in a relatively long conversational context. Therefore, the randomness in the conversation you mentioned can be largely avoided. Also, although topics and talks from different samples (scenes) can be different, character features and traits behind these conversations can be quite consistent. For instance, a character known for humor, like Sheldon, consistently exhibits his comedic nature irrespective of the context—whether on a date or among friends. It's important to note that we construct positive pairs using character representation extracted from the entire conversation, not limited to individual utterances.
>
> __Our contribution__
> Concerning the second rejection reason, we are the first to leverage cross-sample contrastive learning to address the text length issue mentioned in Section 1. We believe this breakthrough is even more critical than designing new contrastive learning pipelines. Additionally, we conduct fine-grained and in-depth analysis in Sections 6.4, 6.5 and 6.6 to validate our cross-sample contrastive learning comprehensively. All these analyses, designs and validation separate our work from only applying existing contrastive learning methods on the character understanding tasks.
>
> __Meaning of "fine-grained"__
> As for the first question, we use fine-grained to refer to the rich and detailed information in conversations. However, as we state in Section 1, lines 47-59, the intricacy of conversation comprehension can hinder PLM from comprehending character well within such complex contexts. Therefore, we introduce summary as an additional and supplemental corpus to conversation, to facilitate PLM to better capture character information from fine-grained conversation. Although summaries may offer a more global and general perspective, they do not impede our utilization of this supplementary information to enhance our understanding of characters within fine-grained conversations.
>
> __Addtional costs__
> Regarding the second question, as we show in Table 8, our additional post-training stage (the first stage) uses less than half of the training epoch numbers of the second before the convergence of the two contrastive losses. In our experiments, the additional time and computational cost are about half of the normal training process.
>
> In conclusion, we sincerely appreciate your meticulous evaluation and kindly request your reconsideration of our paper based on the above response.

---

### Official Review · Reviewer_ydTT · 2023-08-11

**Soundness:** 4

**Excitement:**

4: Strong: This paper deepens the understanding of some phenomenon or lowers the barriers to an existing research direction.

**Missing References:**

None

**Paper Topic And Main Contributions:**

A very interesting problem to solve. The paper tries to understand the characters in scripts which can span a few thousand pages too. The paper uses contrastive learning to solve the problem and achieves reasonable results. The comparison with LLMs is interesting and how the authors are able to beat the large models using summary and conversation encodings, is very interesting.

**Questions For The Authors:**

No questions.

**Reasons To Accept:**

The research is exhaustive, and results are convincing. The intuition of using a mix of summary and conversation data along with contrastive learning is interesting. The research can open avenues for enriching this with visual signals (in case the authors want to extend script analytics to videos etc.) and create more enriched character behavior models which could be of help in many fields.

**Reasons To Reject:**

I would have loved to see how the character understanding will change/enrich with visual signals. A lot of times textual content may not entirely convey the subtle sarcasm etc. Depending on the nature and details in the script some fine grained data elements can get missed. Could be choice of cloths or tone and pitch of conversations. If the authors could have extended the work on those aspects it would have made it more interesting to read.

**Reproducibility:**

3: Could reproduce the results with some difficulty. The settings of parameters are underspecified or subjectively determined; the training/evaluation data are not widely available.

**Reviewer Confidence:**

3: Pretty sure, but there's a chance I missed something. Although I have a good feel for this area in general, I did not carefully check the paper's details, e.g., the math, experimental design, or novelty.

---

> ### Author Rebuttal · Authors · 2023-08-29
>
> We sincerely thank you for acknowledging our efforts in your feedback.
>
> __Expand our work in multi-modal scenarios__
> We also noticed there are many well-built benchmarks and datasets[1][2] in multi-modal narratives. With the recent success of utilizing various contrastive learning methods in the multi-modal area, we are excited to expand our work into these multi-modal areas. Especially considering the key concept of our summary-conversation contrastive learning of leveraging multi-source information, we believe it's promising to apply our method in multi-modal scenarios. Thank you again for your time and effort in evaluating our work!
>
> [1] Kim, Joshua Y., et al. "MONAH: Multi-Modal Narratives for Humans to analyze conversations." Proceedings of the 16th Conference of the European Chapter of the Association for Computational Linguistics: Main Volume. 2021.
>
> [2] Yang, Antoine, et al. "Just ask: Learning to answer questions from millions of narrated videos." Proceedings of the IEEE/CVF international conference on computer vision. 2021.

---

### Official Review · Reviewer_zjYA · 2023-08-12

**Soundness:** 3

**Excitement:**

3: Ambivalent: It has merits (e.g., it reports state-of-the-art results, the idea is nice), but there are key weaknesses (e.g., it describes incremental work), and it can significantly benefit from another round of revision. However, I won't object to accepting it if my co-reviewers champion it.

**Paper Topic And Main Contributions:**

This paper studies the problem of character understanding from show scripts. They show that large language models like ChatGPT perform poorly on this complex task in a zero/few shot setting. To address this, authors propose a new multi-level contrastive learning method to train pre-trained language models (e.g. SpanBERT, $C^2$) in a supervised fashion. On two datasets and three character understanding tasks, the proposed method is shown to outperform relevant baselines.

**Questions For The Authors:**

1.	How is the $Extract$ layer/ method implemented in Equation 4?
2.	How do you determine when the first stage training is completed?
3.	In Table 5, is the hyper-parameter analysis conducted on the test set or the validation set?

**Reasons To Accept:**

1.	The paper proposes a contrastive learning and two stage training method for improving character understanding task with LLMs

2.	Sufficient empirical evaluation is performed to show effectiveness of the proposed approach

**Reasons To Reject:**

1.	The paper presentation and writing can be improved. The character understanding task is not well defined in Section 3 and 4 and hard to follow for unfamiliar readers. It becomes more clearer only in experiments. I suggest giving a concrete example of a specific task at the beginning (e.g. character guessing/ character linking) and using it as a running example to explain the method.
2.	The paper also jumps into the proposed method without providing sufficient intuition as to why this is necessary. The notations are a bit confusing too. As an example, from equation 3, the input sequence is of length $T,$ $(u_1,..., u_T)$ which is encoded by the PLM. As per equation 4, from this PLM encoding $n$ character representations are extracted. How is this extraction performed? No equations are provided to explain this and Figure 1 is unclear. In Section 4, the number of characters is again represented by a different variable $P$
3.	A new two stage training strategy is proposed, but the paper is vague about how to determine when the first stage is completed and the second stage could be started
4.	Experimental results show only a small improvement ~ 1% over baseline. Its not clear if these are statistically significant.

**Reproducibility:**

3: Could reproduce the results with some difficulty. The settings of parameters are underspecified or subjectively determined; the training/evaluation data are not widely available.

**Reviewer Confidence:**

3: Pretty sure, but there's a chance I missed something. Although I have a good feel for this area in general, I did not carefully check the paper's details, e.g., the math, experimental design, or novelty.

**Typos Grammar Style And Presentation Improvements:**

Page 3, line 174: given a certain of text -> given a certain collection/ corpus of text?

Page 4, line 288: post-train -> fine-tune

---

> ### Author Rebuttal · Authors · 2023-08-29
>
> We sincerely thank you for your review of our work and would like to address some points you've raised, including the presentation issue and details of our method.
>
> __Definition to character understanding tasks__
> For the first rejection reason, we acknowledge our work incorporates various script-based character understanding tasks in the experiments. To highlight our novel contrastive losses and enhance the fluency of the methodology section, we use a separate section (Section 5.1) to define each of them in detail. We value your suggestion to add an example in the preceding sections, and will give a more detailed illustration of character understanding in the methodology section.
>
> __Rationale statement in our paper__
> Regarding the second rejection reason, we need to clarify that we give the analysis of the two challenges of script-based character understanding in Section 1, spanning lines 042-072. Subsequently, we explain how the two contrastive losses help to mitigate the aforementioned challenges in lines 073-092. Additionally, we give further intuitive rationales for the two contrastive losses in Section 4.3.1 and 4.3.2 respectively.
>
> __Notation issue__
> Concerning the notation issue, as we mentioned in Section 4.1, lines 203-204, $n$ here represents the number of characters number in the script's conversation. Additionally, in lines 250-251, $P$ is expressly defined as the number of characters that appear in both conversation and summary, which is also the maximum positive pairs we can build in summary-conversation contrastive learning. They just refer to different numbers and here we have $n$>=$P$.
>
> __Character embedding extraction__
> Concerning the extraction equation issue, we use a series of sub-tasks to validate our method and each of them has its own character embedding extraction process. Therefore, we omit the detailed extraction equations to streamline our paper in that part and highlight the two novel contrastive learning losses we proposed. We give the detailed equations below and will include them in the appendix of the next edition of our paper.
>
> __First stage's epoch number__
> As for the third rejection reason, we give detailed epoch numbers for the first and second stages of each single task in Appendix B, Table 8. The specific signal of the epoch number chosen in the first stage is the convergence of the two contrastive losses.
>
> __Our contribution__
> As for the final rejection reason, we would like to underscore the contributions of our work beyond mere quantitative performance enhancements. In this work, we are the first to analyze two challenges in character understanding and address them with novel contrastive losses. In the experiment section, we validate the effectiveness and compatibility of our method with a series of baselines and sub-tasks. In Sections 6.4, 6.5 and 6.6, we also provide fine-grained and in-depth analysis to demonstrate the success of our method in addressing the text type and length challenges in character understanding. Besides, according to our statistics of each model's improvement with our method, we think 1% somehow underestimates our method's performance.
>
> | Model      | Improvemnt |
> |------------|:----------:|
> | SpanBERT   | 1.8%       |
> | C2         | 1.1%       |
> | Bigbird    | 2.7%       |
> | Longformer | 1.6%       |
>
> __Statiscally significance result__
> Below we provide the statistically significance result of our main experiment.  ** denotes that p ≤ 0.01.
>
> | Model                 | B3-PREC | B3-REC  | B3-F1   | CEAFφ4-PREC | CEAFφ4-REC | CEAFφ4-F1 | BLANC-PREC | BLANC-REC | BLANC-F1 | MICRO  | MACRO  |
> |-----------------------|---------|---------|---------|-------------|------------|-----------|------------|-----------|----------|--------|--------|
> | SpanBERT-base         | 77.40   | 82.67** | 79.94   | 74.69       | 67.93      | 71.15**   | 84.80**    | 89.96     | 87.20    | 85.0** | 78.4   |
> | SpanBERT-base (Ours)  | 79.95** | 84.71   | 82.26   | 76.67       | 70.38      | 73.39**   | 87.44      | 91.26     | 89.26    | 86.3   | 78.9** |
> | SpanBERT-large        | 81.92   | 85.56   | 83.69** | 77.85**     | 74.74      | 76.25**   | 88.61**    | 91.91     | 90.20    | 87.2** | 82.8** |
> | SpanBERT-large (Ours) | 83.55** | 87.38** | 85.42** | 79.83       | 76.29      | 78.02**   | 89.18**    | 93.00     | 91.00    | 88.2** | 83.7** |
> | C2-base               | 80.75   | 84.77** | 82.71** | 76.97       | 71.78      | 74.28     | 82.22**    | 91.25     | 89.80**  | 85.6   | 80.4** |
> | C2-base (Ours)        | 83.35   | 85.12** | 84.23   | 76.88**     | 74.97      | 75.91     | 90.48      | 91.85**   | 91.15    | 86.4   | 81.1   |
> | C2-large              | 84.98   | 86.92** | 85.94   | 79.63       | 78.16**    | 78.89     | 90.87**    | 93.05**   | 91.93    | 87.6** | 82.5** |
> | C2-large (Ours)       | 86.42   | 86.44** | 86.24** | 78.82       | 80.42**    | 79.61     | 91.77**    | 93.13     | 92.45**  | 88.0** | 83.2** |
>
> | Model                     | MICRO   | MACRO   |
> |---------------------------|---------|---------|
> | BigBird-P-base            | 71.01   | 70.32** |
> | BigBird-P-base (Ours)     | 72.61   | 73.00   |
> | BigBird-P-large           | 75.43** | 75.24   |
> | BigBird-P-large (Ours)    | 77.68** | 76.41   |
> | Longformer-P-base         | 71.80   | 73.75   |
> | Longformer-P-base (Ours)  | 73.65** | 74.22   |
> | Longformer-P-large        | 77.85   | 75.92** |
> | Longformer-P-large (Ours) | 78.92** | 76.52** |
>
> __Detailed character embedding extraction__
>
> For coreference resolution and character linking, we follow Bai et al 2021.,[1] to initialize the mention-level character embedding:
> \begin{equation}
>     e_i = t_{start_i} + t_{end_i} + e_{speaker_i}
> \end{equation}
> where $t_{start_i}$ and $t_{end_i}$ are the contextualized representation of the beginning and the end tokens of mention $i$, and the $e_{speaker_i}$ is the speaker embedding for the current speaker of the utterance where the $i_{th}$ mention belong to. The speaker embeddings are randomly initialized before training.
>
> For character guessing, we follow Sang et al 2022.,[2] to extract speaker-level character embedding from the context encoding $\mathbf{H}$:
> \begin{equation}
>     A = {\rm Attention}(\mathbf{H})
> \end{equation}
> \begin{equation}
>     a_i = {\rm Softmax}(A \odot M_i)
> \end{equation}
> \begin{equation}
>     e_i = \mathbf{H}^Ta_i
> \end{equation}
>
> where Attention($\cdot$) is a one-layer feedforward network to compute the token-level attention weight. $M_i$ is a token-level mask such that $M_i[j]=1$ if the $j_{th}$ word belongs to an utterance of the $i_{th}$ anonymous speaker and $Mx[j]=0$ otherwise. $a_i$ is the token weight used to pool the hidden states to summarize a character representation.
>
> __Hyper-parameter analysis__
> As for the third question, we conduct hyper-parameter analysis on the test set.
>
> We appreciate other suggestions for our work and believe these presentation-related concerns can be effectively rectified in the subsequent version of our paper. Finally, we kindly request your reconsideration of our paper in light of the aforementioned points.
>
> [1] Bai, Jiaxin, et al. "Joint Coreference Resolution and Character Linking for Multiparty Conversation." Proceedings of the 16th Conference of the European Chapter of the Association for Computational Linguistics: Main Volume. 2021.
>
> [2] Sang, Yisi, et al. "TVShowGuess: Character Comprehension in Stories as Speaker Guessing." Proceedings of the 2022 Conference of the North American Chapter of the Association for Computational Linguistics: Human Language Technologies. 2022.

---

### Meta-Review · Area_Chair_7RMg · 2023-09-25

**Recommendation:** 4

**Metareview:**

The paper proposes a multi-level contrastive learning framework for script-based character understanding. They utilized summary of the script and built a summary-conversation contrastive loss. Experiments are exhaustive and the results are interesting. There are some aspects that need further attention: novelty of the proposed contrastive loss based model; and detailed qualitative explanation.

---

### Decision · Program_Chairs · 2023-10-07

**Decision:**

Accept-Main

**Comment:**

The paper proposes a multi-level contrastive learning framework for script-based character understanding. They utilized summary of the script and built a summary-conversation contrastive loss. Experiments are exhaustive and the results are interesting. There are some aspects that need further attention: novelty of the proposed contrastive loss based model; and detailed qualitative explanation.